# Pennelliiside D, a New Acyl Glucose from *Solanum pennellii* and Chemical Synthesis of Pennelliisides

**DOI:** 10.3390/molecules27123728

**Published:** 2022-06-09

**Authors:** Rishni Masimbula, Hiroto Kobayashi, Tenki Nakashima, Yurika Nambu, Naoki Kitaoka, Hideyuki Matsuura

**Affiliations:** Research Faculty of Agriculture, Hokkaido University, Kita 9 Nishi 9, Kita-ku, Sapporo 060-8589, Japan; rish.masimbula@gmail.com (R.M.); hiryouryoukasai_36@eis.hokudai.ac.jp (H.K.); soccer.ikkyunyukon@gmail.com (T.N.); southpart1192@gmail.com (Y.N.); kitaoka@chem.agr.hokudai.ac.jp (N.K.)

**Keywords:** *Solanum pennellii*, acyl glucose, pennelliisides, trichomes, acyl sugars

## Abstract

Acyl glucoses are a group of specialized metabolites produced by Solanaceae. *Solanum pennellii*, a wild-type tomato plant, produces acyl glucoses in its hair-like epidermal structures known as trichomes. These compounds have been found to be herbicides, microbial growth inhibitors, or allelopathic compounds. However, there are a few reports regarding isolation and investigation of biological activities of acyl glucoses in its pure form due to the difficulty of isolation. Here, we report a new acyl glucose, pennelliiside D, isolated and identified from *S. pennellii*. Its structure was determined by 1D NMR and 2D NMR, together with FD-MS analysis. To clarify the absolute configuration of the acyl moiety of 2-methylbutyryl in the natural compound, two possible isomers were synthesized starting from *β*-D-glucose pentaacetate. By comparing the spectroscopic data of natural and synthesized compounds of isomers, the structure of pennelliiside D was confirmed to be 3,4-*O*-diisobutyryl-2-*O*-((*S*)-2-methylbutyryl)-D-glucose. Pennelliiside D and its constituent fatty acid moiety, (*S*)-2-methylbutanoic acid, did not show root growth-inhibitory activity. Additionally, in this study, chemical synthesis pathways toward pennelliisides A and B were adapted to give 1,6-*O*-dibenzylpennelliisides A and B.

## 1. Introduction

Plants are considered a rich source of natural products that possess diverse structures and corresponding biological activities, such as antiherbivory, antimicrobial, and antioxidant activities [1]. Acyl sugars (sugar esters), nonvolatile secondary metabolites, are specialized natural products produced in the hair-like epidermal structures, known as trichomes, of many Solanaceae families, such as Solanum [1], Nicotina [2], Datura [3], and Petunia [4]. The backbone of acyl sugars basically consists of either a glucose or sucrose moiety attached to one or more straight or branched-chain fatty acids via *O*-acylation [1].

*Solanum pennellii*, a wild tomato species, is endemic to South America [5], and its genomic sequences [6] and introgression lines have been fully characterized [7,8]. It has been reported that *S. pennellii* accumulates various types of secondary metabolites in trichomes, such as terpenoids, phenylpropanoids, and acyl sugars [4,9]. Among them, acyl sugars account for 20% of the total dry weight of leaves [4,10]. Acyl sugars show insecticidal effects against aphids [11], pest repellents [12], and weed growth inhibitory activities [13] as well as allelopathic properties [14]. Most acyl glucoses produced by *S. pennellii* contain mono-, di-, or trifatty acid moieties ranging in carbon number from two to twelve [15,16].

Although some studies have been conducted on the biosynthesis of acyl sugars [17], their full discovery remains unclear because of the availability of vastly diverse acyl sugars [15]. This implies the potential to present considerably diverse acyl glucoses in *S. pennellii* as well. However, there are few reports regarding isolation and investigation of their biological activities in their pure form due to the difficulty of isolation [13,17]. That is because of *α* and *β* anomerization at the C-1 position of the glucose moiety. We recently found that *α* and *β* anomerization can be successfully control by benzylation of hydroxyl groups present at the glucose moiety. Using this strategy, three compounds, 2,3,4-*O*-triisobutyryl-D-glucose, 3-*O*-(8-methylnonanoyl)-2,4-*O*-diisobutyryl-D-glucose, and 3-*O*-decanoyl-2,4-*O*-diisobutyryl-D-glucose, namely, pennelliisides A-C, were reported [14].

As a part of our ongoing research, another new analogue of acyl glucose was identified from *S. pennellii*. To determine the absolute configuration of its fatty acid moiety, total synthesis was carried out. Additionally, in this report, chemical synthesis of previously reported 1,6-*O*-dibenzyl penneliisides A and B are presented. Root-growth inhibitory activity of the newly identified compound and its synthesized compound was also investigated.

## 2. Results and Discussion

### 2.1. Isolation and Identification of Pennelliiside D (**1**)

The aerial parts of 80-day-old *S. pennellii* (1.7 kg) were dipped in EtOH for 30 s, and an extract of epicuticular lipophilic wax was obtained by evaporating the organic solvent under reduced pressure. The extract was partitioned between EtOAc and sat. NaHCO_3_. The extract obtained from the EtOAc layer was roughly purified using silica gel column chromatography to give acyl glucoses, followed by benzylation with 2,4,6-tris(benzyloxy)-1,3,5-triazine (TriBOT) to hinder *α* and *β* anomerization as previously reported [14,18,19]. The obtained benzylated derivatives of acyl glucoses were purified using silica gel column chromatography and HPLC to give dibenzyl pennelliiside D (**2**, 19 mg, Figure 1B).

Compound **2** was obtained as a colorless oil. The molecular formula and molecular weight were found to be C_33_H_44_O_9_ and *m*/*z* 584.2992 [M]^+^ (cal. *m*/*z* 584.2985 [M]^+^), respectively, using HRFD-MS data (Appendix A), indicating that **2** has 12 degrees of unsaturation. Based on ^1^H NMR data, signals at *δ*_H_ 4.38 (d, *J* = 7.6 Hz, 1H, H-1), 5.46 (m, 1H, H-2), 5.48 (m, 1H, H-3), 5.30 (dd, *J* = 10.7, 9.5 Hz, 1H, H-4), 3.42 (m, 1H, H-5), and 3.47 (m, 2H, H-6) were identified as protons related to glucopyranose (Table 1, and Appendix A). The presence of glucopyranose was further confirmed by comparing COSY correlations between the signals at H-1/H-2, H-2/H-3, H-3/H-4, and H-4/H-5 (Figure 2A and Appendix A) together with their corresponding coupling constants (Table 1). Meanwhile, NOESY interactions observed due to the cross-peaks of H-2/H-4 and H-1/H-3/H-5 (Figure 2B and Appendix A), and the signal observed at *δ*_C_ 100.3 in ^13^C NMR (Table 1 and Appendix A) also indicated that the glucose moiety exhibited a *β* anomeric structure.

Next, the resonances in relatively lower field at *δ*_H_ 7.26 (t, *J* = 7.4 Hz, 4H, H-3′, H-7′, H-3″, H-7″), 7.12–7.19 (m, 4H, H-4′, H-6′, H-4″, H-6″), and 7.08 (t, *J* = 7.3 Hz, 2H, H-5′, H-5″) (Table 1 and Appendix A), were identified as resonances corresponding to two benzene rings. Based on the HMBC correlations (Figure 2A), signals appearing at *δ*_H_ 4.75 (d, *J* = 12.2 Hz, 1H, H-1′), 4.45 (d, *J* = 12.2 Hz, 1H, H-1′) and 4.33 (d, *J* = 5.5 Hz, 2H, H-1″) (Appendix A) were identified as methylene protons corresponding to benzylidene attached to the C-1 and C-6 positions. Moreover, the presence of two isobutyryl ester moieties was determined according to the ^1^H NMR and ^13^C NMR spectra and COSY and HMBC correlations (Table 1 and Appendix A), and these moieties were attached to C-3 and C-4 positions in the glucose moiety (Figure 2). Similarly, the 2-methylbutyryl fatty acid moiety attached to C-2 was revealed based on COSY and HMBC correlations, as shown in Figure 2. Therefore, the detailed analysis of 2D NMR data clarified the structure of **2** to be 1,6-*O*-dibenzyl-3,4-*O*-diisobutyryl-2-*O*-(2-methylbutyryl)-*β*-D-glucose (Figure 1). 

To afford **1**, compound **2** was subjected to debenzylation with palladium black under a hydrogen gas atmosphere (Figure 1). Compound **1** was obtained as a colorless oil, and the molecular formula and molecular weight were found to be C_19_H_32_O_9_ and *m*/*z* 405.2133 [M + H]^+^ (cal. *m*/*z* 405.2125 [M + H]^+^), respectively, using HRFD-MS data (Appendix A), indicating that **1** has 4 degrees of unsaturation. Summarized ^1^H NMR and ^13^C NMR data of **1** are shown in Table 2. Although ^1^H NMR, ^13^C NMR, COSY, HSQC, and HMBC (Appendix A) data were complex due to the interference of *α* and *β* anomers, assignment of H and C corresponding to the *α* and *β* anomers of **1** were done partially. Assignments of *α* and *β* anomers are shown in Appendix A. Based on the NMR data, the chemical structure of **1** was determined to be 3,4-*O*-diisobutyryl-2-*O*-(2-methylbutyryl)-D-glucose (Figure 1A), although the absolute configuration of the 2-methylbutyryl fatty acid moiety was still unclear [1,13,17].

### 2.2. Synthesis of Pennelliiside D (**1**)

Two possible isomers of dibenzyl pennelliiside D, 1,6-*O*-dibenzyl-3,4-*O*-diisobutyryl-2-*O*-((*S*)-2-methylbutyryl)-*β*-D-glucose (**2**) and 1,6-*O*-dibenzyl-3,4-*O*-diisobutyryl-2-*O*-((*R*)-2-methylbutyryl)-*β*-D-glucose (**12**), were synthesized to determine the absolute configuration of the fatty acid moiety, 2-methylbutyryl, attached to C-2, although the naturally available ester of 2-methylbutyryl in other natural sources is mostly in the (*S*) configuration [20,21].

Synthesis of **2** was commenced with an available compound, *β*-D-glucose pentaacetate (**3**), by benzylation at C-1 with benzyl alcohol (Figure 2). Removal of acetate groups followed by protection of C-4 and C-6 with benzaldehyde dimethyl acetal and *p*-toluenesulfonic acid resulted in 1-*O*-benzyl-4,6-*O*-benzylidine-*β*-D-glucose (**6**), as reported by Degenstein et al., 2015 [22]. Selective esterification at the C-3 position was achieved by reacting **6** with isobutyryl chloride followed by condensation with (*S*)-2-methylbutanoic acid under a nitrogen gas atmosphere, which offered the desired compound **8.** Cleavage of 4,6-*O*-benzylidine moiety of **8** using triethylsilane and trifluoroacetic acid was done, which was followed by an esterification with isobutyryl chloride to give preferred dibenzyl pennelliiside D (**2**) having (*S*)-configured at the A2 position (Appendix A). Similarly, the synthesis of **12** was achieved starting with **7**, which was conjugated with (*R*)-2-methylbutanoic acid (Appendix A).

Then, we compared the ^1^H NMR and ^13^C NMR data of natural and synthesized compounds (*S*/*R*) (Table 1 and Appendix A). Synthesized (*S*) isomer of dibenzyl pennelliiside D (**2**) had good accordance with natural dibenzyl pennelliiside D (**2**). In the ^1^H NMR, the differences between synthesized (*S*/*R*) with the natural compound were found in the resonances around *δ* 1.65 and *δ* 1.32 as shown in Figure 3. Furthermore, a significant difference was shown when comparing specific rotation values with **12**, while natural **2** and synthesized **2** showed almost the same value. The specific rotation values measured for natural and synthesized **2**, and **12** were [α]^25^_D_ = −10.5, −10.7, and −21.3 (c 0.6, MeOH), respectively. Based on the above observations, we concluded that the absolute configuration of the 2-methylbutyryl fatty acid moiety in natural **2** was (*S*) and confirmed its structure, as shown in Figure 1B.

By debenzylation of synthesized **2** with palladium black under a hydrogen gas atmosphere (Figure 1), synthesized **1** (5.9 mg) was obtained as a colorless oil. The molecular formula and molecular weight were similar to those of the natural **1**, which were C_19_H_32_O_9_ and *m*/*z* 405.2133 [M + H]^+^ (cal. *m*/*z* 405.2125 [M + H]^+^), respectively, using HRFD-MS data (Appendix A). Summarized ^1^H NMR and ^13^C NMR data of synthesized **1** are shown in Table 3. ^1^H NMR, ^13^C NMR, COSY, HSQC, and HMBC data are shown in Appendix A. Similar to natural **1**, partial assignment of H and C corresponding to the *α* and *β* anomers of D-glucose for synthesized **1** is shown in Appendix A. Comparison of ^1^H NMR and ^13^C NMR spectra of natural and synthesized **1** also showed similar data (Table 2 and Table 3, and Appendix A) and revealed the chemical structure of **1** to be 3,4-*O*-diisobutyryl-2-*O*-((*S*)-2-methylbutyryl)-D-glucose (Figure 1A).

### 2.3. Root Growth-Inhibitory Activity of Pennelliiside D (**1**)

Previously, it has been reported that the acyl sucrose showed root growth-inhibitory effect on velvetleaf [23]. Therefore, root growth-inhibitory activity against natural and synthesized **1** and its constituent fatty acid, (*S*)-2-methylbutanoic acid, was assessed. *Arabidopsis thaliana* seeds and 10 µM, 50 µM, and 100 µM concentrations of compounds were used in this experiment. As the control, *A. thaliana* seeds were germinated in the MS medium without adding any compound. The data revealed that neither compound showed root growth-inhibitory activity at any tested concentration (Figure 4), which might support that acyl glucose contains longer chain carbon fatty acids shows root growth-inhibitory effect.

### 2.4. Synthesis of Dibenzyl Pennelliisides A and B

Using the same strategy of synthesis of **2**, synthesis of dibenzyl pennelliisides A and B (**17a, b**) were conducted using **6** as the starting material (Figure 3). Isobutyryl chloride and 8-methylnonanoic acid were used to obtain dibenzyl pennelliisides A and B (**17a, b**). In order to synthesize dibenzyl pennelliiside A, **6** was reacted with isobutyryl chloride to yield **13** that has two isobutyryl fatty acid moieties. Next, deprotection was carried out followed by another reaction with isobutyryl chloride to give the desired compound, dibenzyl pennelliiside A (**17a**). Using the same starting compound, the synthesis of dibenzyl pennelliiside B was commenced with a condensation reaction with 8-methylnonanoic acid to esterify the fatty acid moiety selectively to C-3. Then, **14** was reacted with isobutyryl chloride followed by deprotection and another esterification with isobutyryl chloride to yield dibenzyl pennelliiside B (**17b**). The chemical structures of all compounds were characterized using ^1^H NMR, ^13^C NMR, 2D NMR, and FD-MS (Appendix A). It has been already proven that the removal of benzyl groups can be accomplished as Figure 1 to obtain pennelliisides A and B. Using the same synthesis pathway, it is possible to synthesize other acyl glucoses.

## 3. Materials and Methods

### 3.1. General Experimental Procedures

Optical rotations were obtained with a JASCO P-2200 polarimeter. NMR spectra were recorded in C_6_D_6_, CD_3_OD and CDCl_3_ using a JNM-EX 270 FT-NMR spectrometer (JEOL, ^1^H NMR: 270 MHz) and AMX 500 Bruker system (^1^H NMR: 500 MHz, ^13^C NMR: 126 MHz). Assignment of H and C was performed by obtaining ^1^H NMR, ^13^C NMR (referenced for C_6_D_6_, CD_3_OD and CDCl_3_ at *δ*_H_ 7.16, 3.31 and 7.24, and *δ*_C_ 128.4, 49.2 and 77.2, respectively), COSY, HSQC, HMBC, and NOESY spectra. FD-MS analysis was performed on a JMS-T100GCV (JEOL) instrument. Chromatographic analysis was performed using an HPLC system (InertSustain, A_210max_ nm) equipped with a Shisheido Capcell park C18 column (4.6 × 250 nm, 5 µm, 2 mL/min, MeOH-H_2_O, 80:20) and a Cadenza CK-C18 column (6 × 250 nm, 3 µm, 2 mL/min, MeOH-H_2_O, 80:20). All moisture-sensitive reactions were performed under a nitrogen gas atmosphere. All chemicals used in the study were of analytical grade and purchased from Sigma–Aldrich, Tokyo, Japan, Kanto Chemical Co., Inc, Tokyo, Japan, and Cayman Chemical, Ann Arbor, MI, USA.

### 3.2. Plant Material

Seeds of *S. pennellii* were obtained from the National Bioresource Project (NBRP, Tsukuba). The plants were grown under 16 h of light and 8 h of dark for 80 days at 25 °C in an artificial weather room at the Faculty of Agriculture, Hokkaido University, Hokkaido, Japan.

### 3.3. Extraction and Isolation

To extract acyl sugars from *S. pennellii*, 1.7 kg of aerial parts of plants were used. Pieces of plants were roughly divided into five groups. Each group of plant material was dipped in EtOH (1 L) and shaken for 30 s separately. Then, all the solvent fractions (collectively 5 L) were combined, filtered, and concentrated using a rotary evaporator. The obtained crude material was then extracted into EtOAc (500 mL) with sat. NaHCO_3_ (500 mL) by liquid-liquid extraction. After drying the organic layer with MgSO_4_ and evaporating, the obtained crude material was separated using silica gel column chromatography (MeOH-CHCl_3_-CH_3_COOH, 5:95:0.1). Then, acyl glucoses obtained from separation were subjected to benzylation using TriBOT as mentioned previously [14,18,19]. Briefly, 300 mg of TriBOT and 35 µL of trifluoromethanesulfonic acid (TfOH) were added to a mixture of 4.5 g of obtained crude material in 100 mL of 1,4-dioxane under anhydrous conditions. The reaction mixture was stirred for 16 h at room temperature (RT). After evaporation of the organic solvent, the obtained oil compounds were subjected to silica gel column chromatography (EtOAc-*n*-hexane-CH_3_COOH, 20:80:0.1) to give subfractions. The fraction named Fr2-1 (67.7 mg) was further separated using two consecutive HPLC separations (Shisheido Capcell park C18, 4.6 × 250 nm, 5 µm, 2 mL/min, CH_3_CN-H_2_O, 80:20; Cadenza CK-C18, 6 × 250 nm, 3 µm, 2 mL/min, MeOH-H_2_O, 80:20) to yield colorless oil, **2** (19 mg).

### 3.4. Synthesis of Pennelliiside D (**1**)

Compounds **4**, **5**, **6,** and **9** were synthesized according to a reported method [22].

#### 3.4.1. Synthesis of 1-*O*-Benzyl-2,3,4,6-*O*-tetraacetyl-*β*-D-glucose (**4**)

To a mixture of *β*-D-glucose pentaacetate (**3**, 2.00 g, 5.12 mmol) in 20 mL of anhydrous CH_2_Cl_2_, benzyl alcohol (1.12 mL, 10.25 mmol) and BF_3_·Et_2_O (0.82 mL, 6.66 mmol) were added. The reaction mixture was stirred for 24 h at RT. Then, 10 mL of CH_2_Cl_2_ was added to dilute the reaction mixture. The resulting solution was partitioned between sat. NaHCO_3_ (30 mL × 3) and CH_2_Cl_2_. The organic layer was washed with water (30 mL × 3) and dried over Na_2_SO_4_ followed by evaporation of the organic solvent to result in a crude product. The crude material was purified using silica gel column chromatography (EtOAc-*n*-hexane, 30:70) to yield a colorless oil, **4** (875.5 mg, 2.00 mmol, 39%). ^1^H NMR (270 MHz, CDCl_3_, Appendix A): *δ*_H_ 7.15–7.32 (m, 5H, Ar-H), 4.92–5.12 (m, 3H, H-2, H-3, H-4), 4.54 (d, *J* = 12.3 Hz, 1H, H-7), 4.46 (d, *J* = 7.6 Hz, 1H, H-7), 4.46 (d, *J* = 7.6 Hz, 1H, H-1), 4.19 (dd, *J* = 11.9, 4.67 Hz, 1H, H-6), 4.08 (dd, *J* = 12.7, 2.51 Hz, 1H, H-6), 3.59 (m, 1H, H-5), 2.02 (s, 3H, CH_3_), 1.93 (s, 3H, CH_3_), 1.92 (s, 3H, CH_3_), 1.91 (s, 3H, CH_3_); ^13^C NMR (126 MHz, CDCl_3_, Appendix A): *δ*_C_ 170.7, 170.3, 169.4, 169.3, 136.6, 128.7–128.5, 99.3, 72.9, 71.9, 71.3, 70.8, 68.4, 62.0, 20.7, 20.6 (3C); HRFD-MS *m*/*z* 438.1516 [M]^+^ (calcd for C_21_H_26_O_10_ *m*/*z* 438.1526 [M]^+^) (Appendix A).

#### 3.4.2. Synthesis of 1-*O*-Benzyl-*β*-D-glucose (**5**)

Triethylamine (1.6 mL) and H_2_O (1.6 mL) were added to the reaction mixture containing **4** (875.5 mg, 2.00 mmol) dissolved in MeOH (13 mL). The reaction mixture was stirred at RT for 3 h and concentrated using a rotary evaporator. The resulting residue was purified using silica gel column chromatography (MeOH-CH_2_Cl_2_, 20:80) to give white powder, **5** (529.0 mg, 1.96 mmol, 98%). ^1^H NMR (270 MHz, CD_3_OD, Appendix A): *δ*_H_ 7.15–7.43 (m, 5H, Ar-H), 4.89 (d, *J* = 11.8 Hz, 1H, H-7), 4.62 (d, *J* = 11.8 Hz, 1H, H-7), 4.3 (d, *J* = 7.8 Hz, 1H, H-1), 3.85 (dd, *J* = 11.8, 2.0 Hz, 1H, H-6), 3.64 (dd, *J* = 12.1, 5.4 Hz, 1H, H-6), 3.14–3.33 (m, 4H, H-2, H-3, H-4, H-5); ^13^C NMR (126 MHz, CD_3_OD, Appendix A): *δ*_C_ 137.7, 127.9 (2C), 127.8 (2C), 127.3, 101.9, 76.7, 76.6, 73.8, 70.4, 70.3, 61.4; HRFD-MS *m*/*z* 271.1177 [M + H]^+^ (calcd for C_13_H_18_O_6_ *m*/*z* 271.1182 [M + H]^+^) (Appendix A).

#### 3.4.3. Synthesis of 1-*O*-Benzyl-4,6-*O*-benzylidine-*β*-D-glucose (**6**)

To a mixture of **5** (529.0 mg, 1.96 mmol) and benzaldehyde dimethyl acetal (PhCH(OMe)_2_) (0.35 mL, 2.35 mmol), *p*-toluenesulfonic acid (TsOH·H_2_O) (92.9 mg, 0.49 mmol) dissolved in dimethylformamide (DMF) (5 mL) was added. The reaction mixture was stirred for 5 min at RT, heated to 80 °C, and stirred for 4 h. Then, it was allowed to cool to RT and evaporated using a rotary evaporator. The obtained residue was subjected to liquid-liquid extraction with CH_2_Cl_2_ (20 mL) and sat. NaHCO_3_ (20 mL × 3). The organic layer was collected, dried over Na_2_SO_4_, and evaporated. The obtained crude material was purified with silica gel column chromatography (EtOAc-*n*-hexane, 50:50) to give an oil, **6** (357.7 mg, 1.00 mmol, 51%). ^1^H NMR (500 MHz, CDCl_3_, Appendix A): *δ*_H_ 7.47–7.54 (m, 2H, Ar-H), 7.27–7.40 (m, 8H, Ar-H), 5.51 (s, 1H, H-7), 4.91 (d, *J* = 11.6 Hz, 1H, H-1′), 4.61 (d, *J* = 11.6 Hz, 1H, H-1′), 4.47 (d, *J* = 7.8 Hz, 1H, H-1), 4.34 (dd, *J* = 10.9, 5.0 Hz, 1H, H-6), 3.73–3.82 (m, 2H, H-3, H-6), 3.50–3.57 (m, 2H, H-2, H-4), 3.39–3.46 (m, 1H, H-5); ^13^C NMR (126 MHz, CDCl_3_, Appendix A): *δ*_C_ 136.9, 136.7, 125.9–129.6, 102.1, 101.9, 80.5, 74.5, 73.1, 71.5, 68.6, 66.4; COSY, HSQC, and HMBC data are shown in Appendix A; HRFD-MS *m*/*z* 358.1408 [M]^+^ (calcd for C_20_H_22_O_6_ *m*/*z* 358.1416 [M]^+^) (Appendix A).

#### 3.4.4. Synthesis of 1-*O*-Benzyl-4,6-*O*-benzylidine-3-*O*-isobutyryl-*β*-D-glucose (**7**)

To **6** (357.7 mg, 1.00 mmol) dissolved in anhydrous pyridine (40 mL) at 0 °C, isobutyryl chloride (0.11 mL, 1.00 mmol) was added. The reaction mixture was stirred for 24 h, neutralized with 1 M HCl and evaporated using a rotary evaporator. The obtained crude material was partitioned between EtOAc (50 mL) and 1 M HCl (50 mL × 2) and between EtOAc (50 mL) and sat. NaHCO_3_ (50 mL × 2). The organic layer was washed with H_2_O (50 mL), dried over Na_2_SO_4_, and evaporated to give an oil, which was subjected to silica gel column chromatography (EtOAc-*n*-hexane, 25:75) to yield **7** (149.7 mg, 0.35 mmol, 35%). ^1^H NMR (270 MHz, C_6_D_6_, Appendix A): *δ*_H_ 7.41–7.53 (m, 2H, Ar-H), 6.90–7.21 (m, 8H, Ar-H), 5.34 (dd, *J* = 10.5, 9.48 Hz, 1H, H-3), 5.07 (s, 1H, H-7), 4.64 (d, *J* = 11.8 Hz, 1H, H-1′), 4.25 (d, *J* = 11.8 Hz, 1H, H-1′), 4.12 (d, *J* = 7.6 Hz, 1H, H-1), 4.01 (dd, *J* = 10.3, 4.8 Hz, 1H, H-6), 3.55 (m, 1H, H-6), 3.25–3.42 (m, 2H, H-2, H-4), 2.88–3.20 (m, H, H-5), 2.51 (m, H, H-B2), 0.98 (d, *J* = 4.8 Hz, 3H, CH_3_), 0.95 (d, *J* = 4.5 Hz, 3H, CH_3_); ^13^C NMR (126 MHz, C_6_D_6_, Appendix A): *δ*_C_ 176.5, 137.7, 137.5, 126.1–128.5 (10C), 102.9, 101.1, 78.8, 73.5 (2C), 71.1, 68.5, 66.1, 34.0, 18.9, 18.7; HRFD-MS *m*/*z* 429.1904 [M + H]^+^ (calcd for C_24_H_28_O_7_ *m*/*z* 429.1913 [M + H]^+^) (Appendix A).

#### 3.4.5. Synthesis of 1-*O*-Benzyl-4,6-*O*-benzylidine-3-*O*-isobutyryl-2-*O*-((*S*)-2-methylbutyryl)-*β*-D-glucose (**8**)

Dicyclohexylcarbodiimide (DCC) (294.3 mg, 1.40 mmol) and 4-dimethylaminopyridine (DMAP) (64.7 mg, 0.52 mmol) were added to **7** (149.7 mg, 0.35 mmol). A mixture of (*S*)-2-methylbutanoic acid (0.17 mL, 1.40 mmol) in anhydrous CH_2_Cl_2_ (35 mL) was added to the above mixture, and it was stirred for 24 h at RT. After evaporating volatile components in the reaction mixture, the obtained crude material was subjected to liquid-liquid extraction with CH_2_Cl_2_ (50 mL) and sat. NaHCO_3_ (50 mL × 2), followed by washing the organic layer with 1 M HCl (50 mL × 2) and H_2_O (50 mL × 2). After drying over Na_2_SO_4_ and evaporating, purification was performed using silica gel column chromatography (EtOAc-*n*-hexane, 30:70) to yield a pale green oil, **8** (128.9 mg, 0.25 mmol, 72%). ^1^H NMR (500 MHz, C_6_D_6_, Appendix A): *δ*_H_ 7.56–7.61 (m, 2H, Ar-H), 7.21–7.27 (m, 2H, Ar-H), 7.12–7.18 (m, 4H, Ar-H), 7.06–7.12 (m, 2H, Ar-H), 5.57 (dd, *J* = 10.4, 9.6 Hz, 1H, H-3), 5.47 (dd, *J* = 8.5, 8.0 Hz, 1H, H-2), 5.17 (s, 1H, H-7), 4.74 (d, *J* = 12.1 Hz, 1H, H-1′), 4.37 (d, *J* = 6.6 Hz, 1H, H-1′), 4.35 (d, *J* = 6.9 Hz, 1H, H-1), 4.12 (dd, *J* = 10.0, 5.0 Hz, 1H, H-6), 3.44 (dd, *J* = 12.8, 10.2 Hz, 1H, H-6), 3.38 (dd, *J* = 10.2, 9.4 Hz, 1H, H-4), 3.10 (m, 1H, H-5), 2.47 (m, 1H, H-B2), 2.34 (m, 1H, H-A2), 1.73 (m, 1H, H-A4), 1.34 (m, 1H, H-A4), 1.07–1.15 (m, 9H, H-A3, H-B3, H-B4), 0.84 (t, *J* = 7.4 Hz, 3H, A5); ^13^C NMR (126 MHz, C_6_D_6_, Appendix A): *δ*_C_ 176.1, 175.0, 139.0, 138.3, 126.5–129.8 (10C), 101.8, 101.3, 79.3, 72.4, 72.1, 71.3, 69.0, 66.8, 41.8, 34.6, 27.3, 16.9–19.7 (3C), 12.15; COSY, HSQC, HMBC, and NOESY data are shown in Appendix A; HRFD-MS *m*/*z* 511.2338 [M-H]^+^ (calcd for C_29_H_36_O_8_ *m*/*z* 511.2332 [M-H]^+^) (Appendix A).

#### 3.4.6. Synthesis of 1,6-*O*-Dibenzyl-3-*O*-isobutyryl-2-*O*-((*S*)-2-methylbutyryl)-*β*-D-glucose (**9**)

To **8** (129.0 mg, 0.25 mmol) in anhydrous CH_2_Cl_2_ (10 mL) at 0 °C, trifluoroacetic acid (25. 3 µL, 0.76 mmol) and triethylsilane (Et_3_SiH) (121.7 µL, 0.76 mmol) were added. The reaction was carried out at RT overnight. The reaction mixture was diluted by adding EtOAc (20 mL) and subjected to liquid-liquid extraction with EtOAC (20 mL) and sat. NaHCO_3_ (30 mL × 2), followed by washing the organic layer with 1 M HCl (30 mL) and H_2_O (30 mL). The obtained organic layer was dried over Na_2_SO_4_ and evaporated under reduced pressure. Next, purification was performed by silica gel column chromatography (EtOAc-*n*-hexane, 30:70) to yield an oil, **9** (63.4 mg, 0.12 mmol, 49%). ^1^H NMR (500 MHz, C_6_D_6_, Appendix A): *δ*_H_ 7.04–7.28 (m, 10H, Ar-H), 5.43 (dd, *J* = 10.3, 9.4 Hz, 1H, H-2), 5.23 (dd, *J* = 10.5, 9.2 Hz, 1H, H-3), 4.78 (d, *J* = 12.5 Hz, 1H, H-1′), 4.45 (d, *J* = 12.2 Hz, 1H, H-1′), 4.37 (d, *J* = 8.0 Hz, 1H, H-1), 4.33 (d, *J* = 5.1 Hz, 2H, H-1″), 3.61 (m, 3H, H-4, 2H-6), 3.23 (m, 1H, H-5), 2.44 (m, 1H, H-B2), 2.31 (m, 1H, H-A2), 1.70 (m, 1H, H-A4), 1.34 (m, 1H, H-A4), 1.06–1.12 (m, 9H, H-A3, H-B3, H-B4), 0.83 (t, *J* = 7.4 Hz, 3H, A5); ^13^C NMR (126 MHz, C_6_D_6_, Appendix A): *δ*_C_ 177.5, 174.9, 138.9, 138.1, 128.1–129.1 (10C), 100.5, 76.4, 75.4, 74.0, 71.7, 71.6, 70.8, 70.7, 41.9, 34.6, 27.2, 19.4 (2C), 17.2, 12.2; COSY, HSQC, HMBC, and NOESY data are shown in Appendix A; HRFD-MS *m*/*z* 513.2483 [M-H]^+^ (calcd for C_29_H_38_O_8_ *m*/*z* 513.2488 [M-H]^+^) (Appendix A).

#### 3.4.7. Synthesis of 1,6-*O*-Dibenzyl-3,4-*O*-diisobutyryl-2-*O*-((*S*)-2-methylbutyryl)-*β*-D-glucose (**2**)

To **9** (63.4 mg, 0.12 mmol) dissolved in anhydrous pyridine (10 mL) at 0 °C, isobutyryl chloride (64.7 µL, 0.62 mmol) was added. The reaction mixture was stirred for 24 h, neutralized with 1 M HCl and evaporated using a rotary evaporator. The obtained crude material was partitioned between EtOAc (30 mL) and 1 M HCl (30 mL × 2) and between EtOAc (30 mL) and sat. NaHCO_3_ (30 mL × 2). The organic layer was washed with H_2_O (30 mL), dried over Na_2_SO_4_, and evaporated to give an oil, which was subjected to silica gel column chromatography (EtOAc-*n*-hexane, 20:80) to yield **2** (37.5 mg, 0.06 mmol, 52%). For ^1^H NMR and ^13^C NMR, see Appendix A, and for COSY, HSQC, HMBC, and NOESY data, see Appendix A); HRFD-MS *m*/*z* 584.2995 [M]^+^ (calcd for C_33_H_44_O_9_ *m*/*z* 584.2985 [M]^+^) (Appendix A).

#### 3.4.8. Removal of Benzyl Ether

To a solution of natural **2** (13.3 mg, 0.02 mmol) in 2 mL of EtOAc, 4 mg of palladium black was added. The reaction mixture was stirred for 5 h at RT under a H_2_ gas atmosphere. Then, it was filtered using celite, and volatile components were evaporated under reduced pressure. The obtained crude material was purified using silica gel column chromatography (EtOAc-*n*-hexane, 60:40) to yield an oil, natural **1** (6 mg, 0.01 mmol, 65%). Similarly, synthesized **1** was obtained as a colorless oil (13 mg, 0.03 mmol, 54%) from synthesized **2** (35 mg, 0.06 mmol).

### 3.5. Synthesis of Pennelliisides A and B

#### 3.5.1. Synthesis of 1-*O*-Benzyl-4,6-*O*-benzylidine-2,3-*O*-diisobutyryl-*β*-D-glucose (**13**)

To **6** (123.8 mg, 0.35 mmol) dissolved in anhydrous pyridine (5 mL) at 0 °C, isobutyryl chloride (368 µL, 3.50 mmol) was added. The reaction condition and purification method were similar to the synthesis of **7**. Silica gel column chromatography (EtOAc-*n*-hexane, 15:85) was used to separate **13** (129.0 mg, 0.26 mmol, 75%). ^1^H NMR (500 MHz, C_6_D_6_, Appendix A): *δ*_H_ 7.57 (d, *J* = 7.2 Hz, 2H, Ar-H), 7.23 (d, *J* = 7.5 Hz, 2H, Ar-H), 7.15 (m, 4H, Ar-H), 7.10 (m, 2H, Ar-H), 5.55 (dd, *J* = 9.6, 9.6 Hz, 1H, H-3), 5.45 (dd, *J* = 7.9, 7.9 Hz, 1H, H-2), 5.17 (s, 1H, H-7), 4.73 (d, *J* = 12.3 Hz, 1H, H-1′), 4.36 (d, *J* = 12.5 Hz, 1H, H-1′), 4.43 (d, *J* = 7.9 Hz, 1H, H-1), 4.10 (dd, *J* = 5.0, 5.0 Hz, 1H, H-6), 3.43 (dd, *J* = 10.2 Hz, 1H, H-6), 3.37 (dd, *J* = 9.6, 9.6 Hz, 1H, H-4), 3.07 (m, 1H, H-5), 2.46 (m, 2H, H-A2, B2), 1.04–1.14 (m, 12H, H-A3, A4, B3, B4); ^13^C NMR (126 MHz, C_6_D_6_, Appendix A): *δ*_C_ 175.5, 174.8, 137.6, 137.3, 126.1–128.3 (10C), 101.1, 100.5, 78.5, 72.0, 71.5, 70.6, 68.3, 66.1, 33.9 (2C), 18.8, 18.7 (3C); HRFD-MS *m*/*z* 499.2317 [M + H]^+^ (calcd for C_24_H_28_O_7_ *m*/*z* 499.2332 [M + H]^+^) (Appendix A).

#### 3.5.2. Synthesis of 1-*O*-Benzyl-4,6-*O*-benzylidine-3-*O*-(8-methylnonanoyl)-*β*-D-glucose (**14**)

To a mixture of DCC (58.8 mg, 0.28 mmol) and DMAP (8.7 mg, 0.07 mmol) and **6** (100.0 mg, 0.28 mmol), 8-methylnonanoic acid (17.5 µL, 0.14 mmol) in anhydrous CH_2_Cl_2_ (10 mL) was added. The mixture was stirred for 24 h at RT. After evaporating volatile components in the reaction mixture, the obtained crude material was subjected to liquid-liquid extraction with CH_2_Cl_2_ (50 mL) and sat. NaHCO_3_ (50 mL × 2), followed by washing the organic layer with 1 M HCl (50 mL × 2) and H_2_O (50 mL × 2). After drying over Na_2_SO_4_ and evaporating, purification was performed using silica gel column chromatography (EtOAc-*n*-hexane, 30:70) followed by a preparative TLC (CHCl_3_, 100%) to yield a colorless oil, **14** (48.6 mg, 0.09 mmol, 34%). ^1^H NMR (500 MHz, CDCl_3_, Appendix A): *δ*_H_ 7.46 (m, 2H, Ar-H), 7.27–7.41 (m, 8H, Ar-H), 5.53 (s, 1H, H-7), 5.23 (dd, *J* = 9.4, 9.4 Hz, 1H, H-3), 4.97 (d, *J* = 11.6 Hz, 1H, H-1′), 4.69 (d, *J* = 11.6 Hz, 1H, H-1′), 4.60 (d, *J* = 7.6 Hz, 1H, H-1), 4.41 (dd, *J* = 5.0, 5.0 Hz, 1H, H-6), 3.84 (dd, *J* = 10.2, 10.2 Hz, 1H, H-6), 3.70 (d, *J* = 9.5 Hz, 1H, H-4), 3.66 (dd, *J* = 8.3, 3.9 Hz, 1H, H-2), 3.55 (m, 1H, H-5), 2.53 (s, 1H, H-2OH), 2.40 (t, *J* = 7.4 Hz, 2H, H-B2), 1.65 (m, 2H, H-B3), 1.50 (m, 1H, H-B8), 1.32 (m, 2H, H-B4), 1.29 (m, 2H, H-B5), 1.23 (m, 2H, B6), 1.13 (m, 2H, H-B7), 0.86 (d, *J* = 6.6 Hz, 6H, H-B9, B10); ^13^C NMR (126 MHz, C_6_D_6_, Appendix A): *δ*_C_ 173.9, 136.9, 136.6, 128.0–129.16 (10C), 102.5, 101.5, 78.5, 73.5, 73.3, 71.7, 68.7, 66.6, 38.9, 34.4, 29.5, 29.0, 27.9, 27.2, 25.1, 22.6 (2C); HRFD-MS *m*/*z* 513.2862 [M-H]^+^ (calcd for C_30_H_40_O_7_ *m*/*z* 513.2852 [M-H]^+^) (Appendix A).

#### 3.5.3. Synthesis of 1-*O*-Benzyl-4,6-*O*-benzylidine-2-*O*-isobutyryl-3-*O*-(8-methylnonanoyl)-*β*-D-glucose (**15**)

To **14** (48.6 mg, 0.09 mmol) dissolved in anhydrous pyridine (5 mL) at 0 °C, isobutyryl chloride (50.00 µL, 0.47 mmol) was added, and the reaction mixture was stirred for 24 h. Separation method was same as synthesis of **7**. Compound **15** was obtained as a colorless oil (43.1 mg, 0.07 mmol, 78%). ^1^H NMR (500 MHz, C_6_D_6_, Appendix A): *δ*_H_ 7.58 (m, 2H, Ar-H), 7.05–7.25 (m, 8H, Ar-H), 5.60 (dd, *J* = 9.6, 9.6 Hz, 1H, H-3), 5.46 (dd, *J* = 7.9, 7.9 Hz, 1H, H-2), 5.20 (s, 1H, H-7), 4.74 (d, *J* = 12.3 Hz, 1H, H-1′), 4.38 (d, *J* = 12.3 Hz, 1H, H-1′), 4.37 (d, *J* = 7.9 Hz, 1H, H-1), 4.10 (dd, *J* = 4.9, 5.0 Hz, 1H, H-6), 3.45 (dd, *J* = 4.1, 4.6 Hz, 1H, H-6), 3.42 (d, *J* = 6.1 Hz, 1H-4), 3.12 (m, 1H, H-5), 2.50 (m, 1H, H-A2), 2.23 (m, 2H, H-B2) 1.58 (m, 2H, H-B3) 1.44 (m, 1H, H-B8) 1.10–1.16 (m, 12H, H-B4, B5, B6, A3, A4) 1.07 (m, 2H, H-B7) 0.86 (d, *J* = 6.6, 6H, H-B9, B10); ^13^C NMR (126 MHz, C_6_D_6_, Appendix A): *δ*_C_ 174.9, 172.2, 137.5, 137.3, 126.2–128.8 (10C), 101.3, 100.5, 78.5, 72.1, 71.6, 70.6, 68.3, 66.2, 38.9, 34.0 (2C), 29.5, 29.0, 27.9, 27.2, 24.9, 22.5 (2C), 18.8, 18.7; HRFD-MS *m*/*z* 583.3283 [M + H]^+^ (calcd for C_34_H_46_O_8_ *m*/*z* 583.3271 [M + H]^+^) (Appendix A).

#### 3.5.4. Synthesis of **16a**, **b** and **17a**, **b**

The same synthesis strategies as described above for **9** and **2** were employed for the synthesis of **16a**, **b** and **17a**, **b** respectively.

**16a**; 1,6-*O*-dibenzyl-2,3-*O*-diisobutyryl-*β*-D-glucose, (61.8 mg, 0.12 mmol, 48%), colorless oil, ^1^H NMR (500 MHz, C_6_D_6_ Appendix A): *δ*_H_ 7.04–7.29 (m, 10H, Ar-H), 5.41 (dd, *J* = 8.1, 8.0 Hz, 1H, H-2), 5.22 (dd, *J* = 9.6, 9.6 Hz, 1H, H-3), 5.53 (s, 1H, H-7), 4.77 (d, *J* = 12.4 Hz, 1H, H-1′), 4.46 (d, *J* = 12.4 Hz, 1H, H-1′), 4.37 (d, *J* = 7.8 Hz, 1H, H-1), 4.33 (d, *J* = 4.8 Hz, 1H, H-7), 3.63 (m, 1H, H-4), 3.61 (m, 2H, H-6), 3.22 (m, 1H, H-5), 2.38–2.50 (m, 2H, H-A2, B2), 1.06–1.12 (m, 12H, H-A3, A4, B3, B4); ^13^C NMR (126 MHz, C_6_D_6_, Appendix A): *δ*_C_ 177.1, 175.1, 138.5, 137.8, 127.7–128.7 (10C), 100.0, 76.0, 75.0, 73.6, 71.5, 70.5, 70.3, 34.3 (2C), 19.1 (4C); HRFD-MS *m*/*z* 500.2400 [M]^+^ (calcd for C_28_H_36_O_8_ *m*/*z* 500.2410 [M]^+^) (Appendix A).

**16b**; 1,6-*O*-dibenzyl-2-*O*-isobutyryl-3-*O*-(8-methylnonanoyl)-*β*-D-glucose, (22.5 mg, 0.04 mmol, 52%), colorless oil, ^1^H NMR (500 MHz, C_6_D_6_, Appendix A) *δ*_H_ 7.25 (m, 4H, Ar-H), 7.05–7.18 (m, 6H, Ar-H), 5.42 (dd, *J* = 7.9, 8.0 Hz, 1H, H-2), 5.25 (dd, *J* = 9.5, 9.3 Hz, 1H, H-3), 4.77 (d, *J* = 12.2 Hz, 1H, H-8), 4.46 (d, *J* = 12.3 Hz, 1H, H-8), 4.38 (d, *J* = 8.0 Hz, 1H, H-1), 4.31 (d, *J* = 5.2 Hz, 2H, H-1″), 3.64 (dd, *J* = 9.5, 9.5 Hz, 1H, H-4), 3.59 (dd, *J* = 4.7, 1.6 Hz, 2H, H-6), 3.21 (m, 1H, H-5), 2.57 (s, 1H, H-4OH) 2.49 (m, 1H, H-A2), 2.23 (m, 2H, H-B2), 1.60 (m, 2H, H-B3), 1.46 (m, 1H, H-B8), 1.18 (m, 2H, H-B4), 1.10–1.14 (m, 12H, H-B5, B6, B7, A3, A4), 0.87 (d, *J* = 6.6 Hz, 6H, H-B9, B10); ^13^C NMR (126 MHz, C_6_D_6_, Appendix A): *δ*_C_ 174.8, 173.5, 138.1, 137.5, 127.2–128.4 (10C), 99.7, 75.8, 74.6, 73.3, 71.3, 70.8, 70.1, 70.0, 38.9, 34.1 (2C), 29.5, 29.1, 27.9, 27.2, 24.9, 22.5 (2C), 18.8, 18.7; HRFD-MS *m*/*z* 585.6830 [M + H]^+^ (calcd for C_34_H_48_O_8_ *m*/*z* 584.3349 [M + H]^+^) (Appendix A).

**17a**; dibenzyl pennelliiside A, 1,6-*O*-dibenzyl-2,3,4-*O*-triisobutyryl-*β*-D-glucose, (28.7 mg, 0.05 mmol, 41%), yellow oil, [α]^25^_D_ = +29.1 (c 0.5, CHCl_3_), ^1^H NMR (500 MHz, C_6_D_6_, Appendix A) *δ*_H_ 7.21–7.28 (m, 4H, Ar-H), 7.11–7.19 (m, 4H, Ar-H), 7.07 (t, *J* = 7.2 Hz, 2H, Ar-H), 5.44 (dd, *J* = 8.5, 6.2 Hz, 1H, H-2), 5.43 (dd, *J* = 9.0, 8.4 Hz, 1H, H-3), 5.29 (dd, *J* = 10.0, 9.8 Hz, 1H, H-4), 4.74 (d, *J* = 12.7 Hz, 1H, H-8), 4.44 (d, *J* = 12.4 Hz, 1H, H-8), 4.36 (d, *J* = 7.1 Hz, 1H, H-1), 4.32 (d, *J* = 12.2 Hz, 2H, H-1″), 3.47 (m, 2H, H-6), 3.39 (m, 1H, H-5), 2.41 (m, 1H, H-A2), 2.39 (m, 2H, H-B2, C2), 1.07 (d, *J* = 6.8 Hz, 6H, H-A3, A4), 1.06 (d, *J* = 7.1 Hz, 6H, H-B3, B4), 1.01 (d, *J* = 7.0 Hz, 3H, H-C3), 0.97 (d, *J* = 7.0 Hz, 3H, H-C4); ^13^C NMR (126 MHz, C_6_D_6_, Appendix A): *δ*_C_ 175.5, 174.6, 174.5, 138.3, 137.4, 127.2–128.4 (10C), 99.5, 73.5, 73.2, 72.9, 71.3, 70.0, 69.2 (2C), 33.9 (3C), 18.4–18.9 (6C); COSY, HSQC, and HMBC data are shown in Appendix A; HRFD-MS *m*/*z* 570.2818 [M]^+^) (calcd for C_32_H_42_O_9_ *m*/*z* 570.2829 [M]^+^ (Appendix A).

**17b**; dibenzyl pennelliiside B, 1,6-*O*-dibenzyl-2,4-*O*-diisobutyryl-3-*O*-(8-methylnonanoyl)-*β*-D-glucose, (12.3 mg, 0.02 mmol, 49%), colorless oil, [α]^25^_D_ = −13.3 (c 0.6, CHCl_3_), ^1^H NMR (500 MHz, C_6_D_6_, Appendix A) *δ*_H_ 7.23–7.28 (m, 4H, Ar-H), 7.12–7.18 (m, 4H, Ar-H), 7.07 (t, *J* = 7.6 Hz, 2H, Ar-H), 5.49 (dd, *J* = 9.6, 8.2 Hz, 1H, H-3), 5.46 (dd, *J* = 8.8, 8.5 Hz, 1H, H-2), 5.31 (dd, *J* = 9.5, 8.8 Hz, 1H, H-4), 4.75 (d, *J* = 12.2 Hz, 1H, H-8), 4.45 (d, *J* = 12.3 Hz, 1H, H-8), 4.39 (d, *J* = 7.7 Hz, 1H, H-1), 4.33 (d, *J* = 12.3 Hz, 2H, H-1″), 3.48 (m, 2H, H-6), 3.42 (m, 1H, H-5), 2.46 (m, 1H, H-A2), 2.34 (m, 1H, H-C2), 2.22 (t, *J* = 7.5 Hz, 2H, H-B2), 1.59 (m, 2H, H-B3), 1.47 (m, 1H, H-B8), 1.13–1.23 (m, 6H, 1H-B4, B5, B6), 1.07–1.13 (m, 6H, H-A3, A4), 1.05 (d, *J* = 7.0 Hz, 3H, H-C3), 1.00 (d, *J* = 7.0 Hz, 3H, H-C4), 0.88 (d, *J* = 6.6 Hz, 6H, H-B9, B10); ^13^C NMR (126 MHz, C_6_D_6_, Appendix A): *δ*_C_ 175.0, 174.9, 172.7, 138.6, 137.7, 127.7–128.6 (10C), 99.8, 73.9, 73.6, 73.3, 71.7, 70.3, 69.7, 69.6, 39.3, 34.3, 34,3, 34.2, 29.9, 29.4, 28.3, 27.5, 25.2, 22.8 (2C), 19.0, 18.8; COSY, HSQC, and HMBC data are shown in Appendix A; HRFD-MS *m*/*z* 654.3774 [M]^+^) (calcd for C_38_H_54_O_9_ *m*/*z* 654.3769 [M]^+^ (Appendix A).

### 3.6. Root Growth-Inhibitory Activity

*A. thaliana* seeds were washed to remove damaged and decolorized seeds. They were planted on 1/20 MS medium separately supplemented with natural and synthesized **1**, and (*S*)-2-methylbutanoic acid to reach final concentrations of 10 µM, 50 µM, and 100 µM of each compound. As the control, *A, thaliana* seeds were germinated in the MS medium without adding any compound. Compounds were added to the MS medium after autoclaving. Plates were placed vertically and grown in a 16 h light/8 h dark photoperiod. The root length of each seed was measured after 13 days.

## 4. Conclusions

In conclusion, new acyl glucose, pennelliiside D (**1**), was isolated from *S. pennellii*. Its structure was determined by 1D and 2D NMR, and the absolute configuration of the fatty acid moiety of 2-methylbutyryl was identified as (*S*) by comparing NMR data and specific rotation values between natural and synthesized compounds. The chemical structure of pennelliiside D (**1**) was defined as 3,4-*O*-diisobutyryl-2-*O*-((*S*)-2-methylbutyryl)-D-glucose. Moreover, our data showed that pennelliiside D (**1**) and its constituent fatty acid, (*S*)-2-methylbutanoic acid, did not show root growth-inhibitory activity. Additionally, chemical synthesis pathways for making 1,6-*O*-dibenzyl pennelliisides D were applied to give 1,6-*O*-dibenzyl pennelliisides A and B.

## Data Availability

Data are contained within the article and Appendix A.

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
