# Peer review of "Pennelliiside D, a New Acyl Glucose from Solanum pennellii and Chemical Synthesis of Pennelliisides"

_molecules, 2022, doi:10.3390/molecules27123728_

Round 1
Reviewer 1 Report
This manuscript reports the synthesis of Pennellisides derivatives in order to confirm the structure and absolute configuration of a new acyl glucose from Solanum pennellii called Pennelliside D.
The methodology for the synthesis of these compounds is based on published synthetic strategies for the synthesis of this kind of compounds and the transformation are simple but versatile.
The root-growth-inhibitory activity of penneliside D was evaluated but it did not show any inhibitory activity at 100 µM and lower.
The manuscript could be accepted after minor revisions as followings.
1) The chemical shift for the 13C NMR data should be indicated with just one decimal.
2) 13C NMR data for some compounds are missed. For example, 13C NMR data for compounds 4, 5, 7, 13, 14, 15, 16a, 16b, etc. These data should be included or authors should explain the reason why these data are not available.
3) In Table 1, the NMR data for natural and synthesized dibenzyl penneliside D are identical. Authors should simplify the table and remove these duplicate data.
4) Page 9 Line 218, 219 and 221 CH2CH2 should be CH2Cl2
Author Response
Thank you very much for reviewing our manuscript and providing a lot of useful comments and suggestions. We would like to answer your questions and have revised the manuscript as attached.

Reviewer 2 Report
This a nice work on isolation of a new acyl glucose from Solanum pennellii via benzylation strategy. Determine the absolute configuration of the acyl moiety of 2-methylbutyryl in this compound is changeling, which was solved perfectly by synthesizing a pair of isomers.
After reading, I have some concerns about this manuscript.
P2 L62: Does ‘30 s’ mean ’30 seconds’ ? If yes, then there is a question. As shown in Extraction and isolation part, 5 L EtOH were poured into the 1.7 kg plant material, and this cost more than 30 seconds. In other words, the rinsing time for the upper layer should be more than 30 seconds. How to judge the rinsing time is 30 seconds for all the material?
What’s the control in root growth inhibitory bioassay?
As shown, there are two synthetic routines in Scheme 3, but it was not described in detail in the above paragraph. In fact, the upper one is to synthesize dibenzyl pennelliiside A, while the lower one is to synthesize dibenzyl pennelliiside B. Please give a clear description for the synthetic routines, otherwise the readers might be confused.
Aside, some improvements are required for this manuscript. Some are listed as following.
1. Page 3 Paragraph 4: If the integration for a proton peak is one, it is usually recorded as ‘1H’ not ‘H’. Please add the unit ‘Hz’ for the coupling constants. And one coupling constant was missing for ‘5.30 (dd, J4,5 = 10.7, H, H-4)’. Please revise them as well as those in the following Page 4 Paragraph 1 and the NMR data of the synthetic compounds in the Materials and Methods section. In addition, please keep one decimal for the coupling constants of the synthetic compounds 4–8.
2. Table 1: To keep consistent, please keep two decimals for ‘128.1’. And give specific chemical shifts for ‘7.23-7.30, t (7.3)’ and ‘7.05-7.10, t (7.3)’ as each of them is a triplet as shown in Figure S42.
Author Response
Thank you very much for reviewing our manuscript and providing a lot of useful comments and suggestions. We would like to answer your questions and have revised the manuscript.

Round 2
Reviewer 2 Report
The comments has been addressed and the paper is revised accordingly. No further suggestion from me. Now it is a nice paper, and I recommend it for publication.